# Distributional MIPLIB: a Multi-Domain Library for Advancing ML-Guided MILP Methods

**Weimin Huang, Taoan Huang**
University of Southern California

**Aaron Ferber**
Cornell University

**Bistra Dilkina**
University of Southern California

## Abstract

Mixed Integer Linear Programming (MILP) is a fundamental tool for modeling combinatorial optimization problems. Recently, a growing body of research has used machine learning to accelerate MILP solving. Despite the increasing popularity of this approach, there is a lack of a common repository that provides distributions of similar MILP instances across different domains, at different hardness levels, with standardized test sets. In this paper, we introduce *Distributional MIPLIB*, a multi-domain library of problem distributions for advancing ML-guided MILP methods. We curate MILP distributions from existing work in this area as well as real-world problems that have not been used, and classify them into different hardness levels. It will facilitate research in this area by enabling comprehensive evaluation on diverse and realistic domains. We empirically illustrate the benefits of using *Distributional MIPLIB* as a research vehicle in two ways. We evaluate the performance of ML-guided variable branching on previously unused distributions to identify potential areas for improvement. Moreover, we propose to learn branching policies from a mix of distributions, demonstrating that mixed distributions achieve better performance compared to homogeneous distributions when there is limited data and generalize well to larger instances.

## 1 Introduction

Mixed Integer Linear Programming (MILP) is an essential technique for modeling and solving Combinatorial Optimization (CO) problems, covering a wide range of applications such as production planning and scheduling [83]. Many CO problems are NP-complete or NP-hard [61, 31, 51, 43, 86] and are thus inherently challenging to solve. Exact algorithms [79] and heuristics [72, 8] have been studied for MILPs. However, solving MILPs remains challenging as problems scale in size and complexity, coupled with the increasing demand for real-time solutions.

Many algorithmic decisions in exact and heuristic algorithms for MILPs traditionally rely on intuition from problem structures and/or are manually made based on evaluation on specific instances [68, 59]. However, manual tuning requires domain-specific knowledge and may fail to realize the full performance potential of algorithms. In recent years, Machine Learning (ML) has been proposed to address this shortcoming. There has been an increasing interest in enhancing MILP-solving frameworks with adaptable learning components that exploit the correlation between algorithmic patterns and the performance of the algorithm [22, 25]. For example, [46, 26, 33, 85, 37, 52] improves Branch-and-Bound (B&B), a tree search algorithm used in MILP solvers [35, 14], with ML.

Despite the increasing popularity of ML-guided MILP solving, there is a lack of a common repository containing distributions of MILPs along with standardized test sets for ML approaches. Some researchers [63, 64] use MIPLIB, a library containing various MILP instances that differ in difficulties, structures, and sizes [29]. While MIPLIB has been traditionally used for benchmarking MILP solvers, its instances are heterogeneous, making it less suited for ML-based methods. As ML typically benefits from a large amount of data from a certain distribution, it remains a challenge for ML methods to

Submitted to 39th Conference on Neural Information Processing Systems (NeurIPS 2025). Do not distribute.

deliver state-of-the-art performance on MIPLIB instances [63, 64]. To leverage data in distributional settings, much of the existing work independently generates MILP distributions. This leads to two issues. First, the lack of standardized test sets makes it hard to benchmark and compare between different methods. Second, a small set of synthetic domains has been repeatedly used and there is a lack of evaluation on real-world domains, making the evaluation not comprehensive. In Table 1, we summarize problem domains used in representative papers for different learning tasks. Classical problems such as Set Covering (SC), Combinatorial Auction (CA), Maximum Independent Set (MIS), and Capacitated Facility Location Problem (CFLP) are commonly used. Although these problems are NP-hard, the instances are synthetic and less challenging compared to many real-world problems. A few papers [64, 75, 76] have used instances from real-world problems such as the production packing problem and electric grid optimization [64], but few of the real-world datasets are publicly accessible, making it hard to reproduce the results.

Table 1: MILP distributions used in previous work. [†] indicates those that are not publicly available. For each work, we mark whether the ML component is trained on distributions of Single Domain (SD), Mixed Domain (MD), or MIPLIB. We also mark whether it is tested in problem domains that are the same as training (ID), in the same domain but on larger distributions (ID(L)), or out of domains (OD). The names of domains corresponding to the abbreviations are in Sec 3.1 and Appendix A.

| Track | Paper | Training | Testing | Problem Domains |
|---|---|---|---|---|
| **Learning for B&B** | | | | |
| Branching | Khalil et al. [46] | SD | ID | MIPLIB |
| | Gasse et al. [26] | SD | ID(L) | SC, CA, CFLP, MIS |
| | Nair et al. [64] | SD + MIPLIB | ID + MIPLIB | CORLAT, NNV, Google Production Packing[†], Electric Grid Optimization[†], MIPLIB |
| | Gupta et al. [33] | SD | ID(L) | SC, CA, CFLP, MIS |
| | Zarpellon et al. [85] | MIPLIB | MIPLIB | MIPLIB |
| | Gupta et al. [34] | SD | ID(L) | SC, CA, CFLP, MIS |
| | Scavuzzo et al. [74] | SD | ID(L) | SC, CA, CFLP, MIS, MK |
| | Lin et al. [56] | SD | ID(L) | SC, CA, CFLP, MIS |
| Backdoor prediction | Ferber et al. [24] | SD | ID | NNV, CFLP, GISP |
| | Cai et al. [15] | SD | ID(L) | SC, CA, CFLP, MIS, GISP, NNV |
| Node selection | He et al. [37] | SD | ID + OD | CA, CORLAT, MK |
| | Labassi et al. [52] | SD | ID(L) | GISP, Fixed Charge Network Flow, MAXSAT |
| Cut selection | Tang et al. [77] | SD | ID(L) + OD | Packing, Production Planning, Binary Packing, MC |
| | Huang et al. [44] | SD | ID | SC, MK, Production Planning[†] |
| | Li et al. [55] | SD + MIPLIB | ID + MIPLIB | CA, CFLP, MIS, Packing, Binary Packing, MC |
| Run heuristics | Khalil et al. [47] | SD + MIPLIB | ID + MIPLIB | GISP, MIPLIB |
| | Chmiela et al. [17] | SD | ID(L) | GISP, Fixed Charge Network Flow |
| **Learning for meta heuristics** | | | | |
| LNS | Song et al. [75] | SD | ID | CA, MVC, MC, Risk-Aware Path Planning[†] |
| | Wu et al. [84] | SD + MIPLIB | ID(L) + MIPLIB | SC, CA, MIS, MC, MIPLIB |
| | Sonnerat et al. [76] | SD + MIPLIB | ID + MIPLIB | NNV, Google Production Packing[†], Electric Grid Optimization[†], MIPLIB, Google Production Planning[†] |
| | Liu et al. [57] | SD + MIPLIB + MD | MIPLIB + OD | SC, CA, MIS, GISP, MIPLIB |
| | Huang et al. [41] | SD | ID(L) | SC, CA, MIS, MVC |
| Solution prediction | Ding et al. [22] | SD | ID(L) | SC, CFLP, MIS, MK, Fixed Charge Network Flow, TSP, VRP, Generalized Assignment |
| | Nair et al. [64] | SD + MIPLIB | ID + MIPLIB | CORLAT, NNV, Google Production Packing[†], Electric Grid Optimization[†], MIPLIB |
| | Khalil et al. [48] | SD | ID | GISP, Fixed Charge Network Flow |
| | Han et al. [36] | SD | ID(L) | CA, MIS, IP, LB |
| | Huang et al. [42] | SD | ID(L) | CA, MIS, MVC, IP |

This paper introduces *Distributional MIPLIB* (d-MIPLIB), a comprehensive, multi-purpose MILP library encompassing various MILP problem distributions to support the development of ML-guided MILP-solving methods. In this context, a distribution refers to MILPs of the same problem formulation constructed from data parameters sampled from a given distribution. We curate distributions from ten synthetic and real-world problems used in the existing literature on ML for MILPs and three real-world problems for which no ML methods have been attempted. For each problem, distributions are classified into multiple hardness levels. 100 test instances are pre-generated for each distribution, and 900 are pre-generated for training and validation [1]. Additionally, a generator is provided for most problems to generate additional instances for training.

---

[1] The number of test instances in 3 distributions is less than 100 due to limited available data.

*Distributional MIPLIB* will significantly accelerate research in MILP solving and data-driven algorithm design by providing distributional data for ML-based methods and enabling benchmarking. The set of distributions covers various hardness levels and a diverse set of application domains, making it suitable for different types of MILP algorithms (e.g., smaller problems for exact solving and larger problems for heuristic methods). Moreover, the standardized benchmark sets that d-MIPLIB provides not only enable better comparison analysis of different methods, but also enable evaluation across broader domains at different problem scales and on realistic problems, allowing researchers to identify gaps and open up new avenues of novel research.

To demonstrate the potential of *Distributional MIPLIB* in facilitating research, we evaluate the performance of ML-guided variable branching [26] on previously unused distributions and uncover potential areas for improvement. Moreover, we propose to learn branching policies in B&B from a mix of distributions, demonstrating that mixed distributions achieve better performance compared to homogeneous distributions when there is limited data and generalize better to larger instances. Furthermore, we propose several additional directions for utilizing the dataset, highlighting its potential for opening up new research avenues. To encourage further research and facilitate the curation of future distributions, we provide a website for *Distributional MIPLIB*. The URL to the website will be included in the final version of the paper.

## 2 Background and Related Work

Formally, a MILP with $n$ decision variables and $m$ constraints is defined by a coefficient matrix $A \in \mathbb{R}^{m \times n}$, a vector $b \in \mathbb{R}^m$, a cost vector $c \in \mathbb{R}^n$, and a partition $(B, I, C)$ of variables. $B, I, C$ are the sets of indices of binary, general integer, and continuous variables, respectively. The goal is to find $x$ such that $c^T x$ is maximized, subject to linear constraints $Ax \leq b$ and integrality constraints on binary decision variables $x_j \in \{0, 1\}, \forall j \in B$ and integer decision variables $x_j \in \mathbb{Z}, \forall j \in I$.

MILP solvers such as Gurobi [35] and SCIP [14] use Branch-and-Bound (B&B), an exact tree search algorithm, as the core component. B&B starts with the root node representing the original input MILP. It then repeatedly chooses a leaf node and creates two smaller subproblems by splitting the domain of a variable. This step is referred to as *branching*. Besides B&B, meta-heuristics, such as Large Neighborhood Search (LNS) and Predict-and-Search (PaS), are also popular MILP search algorithms that can find high-quality solutions to MILPs much faster than B&B for hard problems, but do not provide optimality guarantees.

### 2.1 Machine Learning for MILP Solving

ML has been proposed to accelerate MILP solving in different ways. A large body of research improves B&B by learning to select which variables to branch on [46, 26, 33, 85] or which nodes to expand in the search tree [37, 52]. There are also works on learning to schedule or execute primal heuristics [47, 17] and to select cutting planes [77, 66, 44] in B&B. ML has also been applied to improve meta-heuristics. [75, 76, 84, 41] apply learning techniques, such as imitation, reinforcement, and contrastive learning, to learn to select which subset of variables to reoptimize in LNS. [64, 36, 42] focus on PaS, where they learn to predict the optimal assignment for part of the variables to get a reduced-size MILP that is easier to solve. A comprehensive literature review is provided Appendix B.

### 2.2 Existing Libraries and Software Packages

MIPLIB [29] is a library that provides access to heterogeneous real-world MILP instances, containing 1065 instances from various domains that are diverse in size, structure, and hardness. It has become a standard test set used to compare the performance of MILP solvers, and several ML methods for MILP solving have been tested on MIPLIB instances [76, 84, 55]. Despite some early success, it remains a challenge for ML methods to deliver state-of-the-art performance on MIPLIB instances, due to the heterogeneous nature [63, 64].

There are also a few open-source software packages built to facilitate research in ML-guided CO. MIPLearn [73] is a software for ML-guided MILP solving that provides access to a complete ML pipeline, including data collection, training, and testing. MIPLearn provides generators for one real-world problem, which is a simplified formulation for Unit Commitment (UC). However, domain knowledge is required to generate realistic UC instances that are more complex and more challenging

to solve. Ecole [71] is a library designed to facilitate research on using ML to improve CO solvers. It exposes the sequential decision-making processes in MILP-solving as control problems over Markov Decision Processes. Currently, Ecole provides instance generators for four classical problems. OR-Gym [45] is a framework for developing RL algorithms to produce high-quality solutions for CO, without using MILP solvers. Ecole and OR-Gym are designed for augmenting solvers and finding high-quality solutions without solvers, respectively. MIPLearn supports both tasks.

**Comparison with Existing Datasets and Contributions.** *Distributional MIPLIB* provides a counterpart for the MIPLIB that provides distributions of similar MILP instances of the same problem formulation, intended for the development and evaluation of ML-guided MILP methods. Similar to MIPLIB, it covers a broad range of application domains, allowing for a more comprehensive evaluation on problems with different structures, especially in real-world domains. All the instances are pre-compiled, and no domain knowledge is required to access complex real-world instances included in d-MIPLIB. It covers multiple hardness levels, making it suitable for a wide range of MILP methods (e.g., exact solving for easier instances and meta-heuristics for harder instances), compared to libraries designed for specific avenues.

## 3 Distributional MIPLIB

We pre-generate MILP distributions from both synthetic and real-world problem domains, classifying them into different hardness levels. Table 2 shows the sources where the distributions were initially used in existing work on ML for MILPs, along with instance statistics. While we pre-compile a fixed number of instances for consistency in comparing methods, an instance generator is available for generating additional training instances for all synthetic problems and one real-world problem (Optimal Transmission Switching).

### 3.1 Data Sources

We curate synthetic instances from domains that have been used in multiple ML tasks in existing work (Table 1). Among variants of a problem in synthetic domains, we choose the most representative one among the variants. As for real-world domains, as many problems used in existing work are proprietary and not publicly available, we cover the associated domain by including an available problem from the same area (e.g., Optimal Transmission Switching as a surrogate for Electric Grid Optimization used in [76]).

**Synthetic Problems.** We curate synthetic instances from domains commonly used in the literature on ML for MILPs. As shown in Table 1, the most frequently used NP-hard problem domains are Combinatorial Auctions (CA) [53], Set Covering (SC) [5], Maximum Independent Set (MIS) [7], Capacitated Facility Location Problem (CFLP) [20], and Minimum Vertex Cover (MVC) [23]. Additionally, we compile distributions of the Generalized Independent Set Problem (GISP), a graph optimization problem proposed for forestry management [39, 18]. We used the instance generators provided in the existing work to compile MILP distributions as described in their work and generate additional distributions covering different hardness levels for frequently used domains such as Minimum Vertex Cover (MVC). Finally, we include Item Placement (IP), which involves spreading items across containers to utilize them evenly [62], and Load Balancing (LB) [82], which deals with apportioning workloads across workers, used in the NeurIPS 2021 Machine Learning for Combinatorial Optimization Competition (ML4CO) [25].

**Real-world Problems.** In addition to synthetic instances, we include MILP instances from five real-world domains. The Maritime Inventory Routing Problem (MIRP) [65] determines routes from production ports to consumption ports to minimize transportation costs and manages the inventory at these ports, covering both ship routing and inventory management. MIRP was used as a hidden test set in ML4CO [25]. Neural Network Verification (NNV) is an optimization problem in ML that verifies the robustness of a neural network on a given input example [16, 78]. The NNV instances we include were derived from verifying a convolutional network on MNIST examples, which was used in [64] for learning for branching and solution prediction.

Furthermore, we compile distributions from problems where no ML method has been applied, covering applications in energy, e-commerce, and sustainability. In energy planning, the Optimal Transmission Switching (OTS) problem under high wildfire ignition risk [70] is a type of Network

Topology Optimization problem. Transmission grids are represented as a series of buses (vertices) connected by power lines (edges). During high wildfire ignition risk, transmission lines can start wildfires; methods to mitigate this risk include de-energizing and undergrounding transmission lines. De-energizing lines prevent fires but interrupt power delivery to customers, whereas undergrounding lines can deliver power without the risk of igniting a fire but at a higher cost. OTS examines the optimal way to de-energize and underground transmission lines to reduce wildfire risk while minimizing power outages within a resource budget. In e-commerce, the Middle-Mile Consolidation Network (MMCN) [31] problem is a network design problem that creates load consolidation plans to transport shipments from stocking locations, including vendors and fulfillment centers, to last-mile delivery locations. It determines a minimum-cost allocation of transportation capacity on network arcs that satisfies shipment lead-time constraints. For MMCN, we include distributions containing binary and integer variables (denoted as BI) and distributions containing binary and continuous variables (denoted as BC) [2]. The Seismic-Resilient Pipe Network (SRPN) Planning [40] is another network design problem in infrastructure resilience space that chooses which water pipes are to be upgraded in earthquake hazard zones to ensure water delivery to critical customers and households during disasters, while minimizing the cost. The SRPN instances in this library are generated based on earthquake hazard zones in Los Angeles.

As shown in Table 2, most synthetic MILP instances contain only binary decision variables, except for CFLP, LB, and IP, which include continuous variables. The real-world problems, on the other hand, encompass diverse distributions with integer and continuous decision variables, enabling comprehensive benchmarking on more realistic and complex problems.

## 3.2 Evaluation

**Instances Data Generation.** For each synthetic distribution, we generate a total of 1000 instances, with 900 intended for training and validation in ML-guided methods and 100 for testing and evaluation. For the real-world problems OTS and MMCN, we follow the same practice as the synthetic problems, providing 100 test instances for each distribution. For NNV, since precompiled train, validation, and test splits are publicly available, we respect the established splits, including the same 588 instances in the test set. However, for MIRP and SRPN, the number of test instances is less than 100 as the total number of instances available is limited. MIRP contains 20 test instances. SRPN contains 22 and 20 test instances in the Easy and Hard group, respectively [3].

**Performance Metrics and Problem Instance Statistics.** We design a set of evaluation metrics that characterize performance well from easy to hard settings. We report the number of instances in the test set that are solved to optimality in 1 hour (# Opt). For instances solved to optimality, we report the average solving time in seconds (Opt Time). For instances not solved to optimality, we report the average primal-dual gap after 1 hour (NonOpt Gap). The primal-dual gap represents the gap between the lower and upper objective bounds. Specifically, let $z_P$ be the primal objective bound (i.e., the value of the best feasible solution found so far, serving as the upper bound for minimization problems), and $z_D$ be the dual objective bound (i.e., linear relaxation of the MILP, serving as the lower bound for minimization problems). The primal-dual gap is defined as $gap = |z_P - z_D|/|z_P|$ [35]. Additionally, we report the primal-dual integral (Integral), which is defined as the integral of the primal-dual gap over time [9], with lower values indicating faster (better) convergence. For instance statistics, we report the average number of binary (# Var B), integer (# Var I), and continuous (# Var C) variables, and the average number of constraints (# Constr).

**Hardness Levels.** We classify distributions into 5 hardness levels based on the runtime statistics. For distributions with at least some instances solved to optimality within 1 hour, we classify them into three levels based on the average solving time. Distributions with average solving times under 100 seconds are categorized as *Easy*, 100-1000 seconds as *Medium*, and those exceeding 1000 seconds as *Hard*. For distributions with no instances solved to optimality within 1 hour, we further classify them into *Very hard* and *Extremely hard* based on the primal-dual gap. *Very hard* and *Extremely hard* (Ext hard) distributions are groups where the primal-dual gap is less than 1 and greater than 1, respectively.

---

[2]BI and BC distributions correspond to 2 variants of MMCN. In the BI variant, all arcs in the network have the same transit mode. In the BC variant, multiple transit modes are allowed.

[3]As MIRP has been used in ML4CO, we adhere to the train, validation, and test split established by ML4CO. For SRPN, we randomly selected 10% of the total instances as the test set for each distribution.

Table 2: Synthetic and Real-world problems in *Distributional* MIPLIB. † indicates domains for which generators are available. ‡ indicates distributions where # test instances is not 100. For the performance metrics, we use Gurobi (v10.0.3) [35] with 1 hour time limit, on a cluster with Intel Xeon Silver 4116 CPUs @ 2.10GHz, with a RAM allocation of 5G (For SRPN-Hard, MMCN-Very Hard, and OTS-Hard instances, we increased the RAM to 15G, due to memory errors at 5GB.)

| Domain | Hardness Level | Dist. Source: ML4MILPs | Instance Statistics | | | | Performance Metrics | | | |
|---|---|---|---|---|---|---|---|---|---|---|
| | | | # Var B | # Var I | # Var C | # Constr | # Opt | Opt Time(s) | NonOpt Gap | Integral |
| *Synthetic* | | | | | | | | | | |
| CA† | Easy | Gasse et al. [26] | 1000 | 0 | 0 | 385.04 | 100 | 47.14 | N/A | 2.30 |
| | Medium | Gasse et al. [26] | 1500 | 0 | 0 | 578.07 | 100 | 358.14 | N/A | 7.29 |
| | Very hard | Huang et al. [41] | 4000 | 0 | 0 | 2676.32 | 0 | N/A | 0.10 | 400.28 |
| SC† | Easy | Gasse et al. [26] | 1000 | 0 | 0 | 500 | 100 | 18.05 | N/A | 0.99 |
| | Medium | Gasse et al. [26] | 1000 | 0 | 0 | 1000 | 100 | 214.11 | N/A | 15.78 |
| | Hard | Gasse et al. [26] | 1000 | 0 | 0 | 2000 | 56 | 1603.66 | 0.04 | 180.25 |
| | Very hard | Huang et al. [41] | 4000 | 0 | 0 | 5000 | 0 | N/A | 0.20 | 847.11 |
| MIS† | Easy | Gasse et al. [26] | 1000 | 0 | 0 | 3946.25 | 100 | 50.52 | N/A | 0.86 |
| | Medium | Gasse et al. [26] | 1500 | 0 | 0 | 5941.14 | 88 | 470.44 | 0.01 | 11.28 |
| | Very hard | Huang et al. [41] | 6000 | 0 | 0 | 23994.82 | 0 | N/A | 0.30 | 1132.69 |
| MVC† | Easy | New | 1200 | 0 | 0 | 5975 | 100 | 27.26 | N/A | 0.27 |
| | Medium | New | 2000 | 0 | 0 | 9975 | 97 | 244.11 | 0.01 | 2.28 |
| | Hard | New | 500 | 0 | 0 | 30100 | 55 | 1821.04 | 0.02 | 102.74 |
| | Very hard | Huang et al. [41] | 1000 | 0 | 0 | 65100 | 0 | N/A | 0.12 | 454.02 |
| GISP† | Easy | New | 605.81 | 0 | 0 | 1967.05 | 100 | 43.09 | N/A | 15.59 |
| | Medium | Ferber et al. [24] | 988.81 | 0 | 0 | 3353.03 | 100 | 671.89 | N/A | 204.83 |
| | Hard | Ferber et al. [24] | 1317.03 | 0 | 0 | 4567.83 | 85 | 2623.16 | 0.08 | 866.16 |
| | Very hard | Cai et al. [15] | 6017 | 0 | 0 | 7821.87 | 0 | N/A | 0.44 | 2104.04 |
| | Ext hard | Khalil et al. [47] | 12675.83 | 0 | 0 | 16515.44 | 0 | N/A | 2.01 | 8139.33 |
| CFLP† | Easy | Gasse et al. [26] | 100 | 0 | 10000 | 10201 | 100 | 44.44 | N/A | 0.57 |
| | Medium | Gasse et al. [26] | 200 | 0 | 20000 | 20301 | 100 | 103.51 | N/A | 0.88 |
| LB † | Hard | Gasse et al. [25] | 1000 | 0 | 60000 | 64307.17 | 9 | 2665.11 | 0.00 | 33.48 |
| IP † | Very hard | Gasse et al. [25] | 1050 | 0 | 33 | 195 | 0 | N/A | 0.44 | 1770.42 |
| *Real-world* | | | | | | | | | | |
| MIRP | Medium | Gasse et al. [25] | 0 | 15080.57 | 19576.15 | 44429.70 | 10‡ | 697.24 | 0.23 | 728.75 |
| NNV | Easy | Nair et al. [64] | 171.49 | 0 | 6972.60 | 6533.70 | 588‡ | 37.98 | N/A | 21.81 |
| OTS† | Easy | New | 4181 | 0 | 17137 | 48582 | 100 | 45.86 | N/A | 3.72 |
| | Medium | New | 7525 | 0 | 33202 | 92992 | 100 | 419.55 | N/A | 25.80 |
| | Hard | New | 6546 | 0 | 46423 | 111804 | 52 | 2564.00 | 0.20 | 1926.19 |
| MMCN | Medium$^{BI}$ | New | 1156.94 | 263.23 | 0 | 437.81 | 100 | 114.93 | N/A | 3.01 |
| | Medium$^{BC}$ | New | 4271.59 | 0 | 324.04 | 3171.23 | 100 | 468.17 | N/A | 37.30 |
| | Hard$^{BI}$ | New | 2074.76 | 346.39 | 0 | 642.57 | 34 | 1998.57 | 0.01 | 79.79 |
| | Very hard$^{BI}$ | New | 21596.72 | 1127.29 | 0.00 | 3944.01 | 0 | N/A | 0.10 | 369.15 |
| | Very hard$^{BC}$ | New | 68345.21 | 0 | 2425.87 | 96272.60 | 0 | N/A | 0.61 | 2761.52 |
| SRPN | Easy | New | 3016.42 | 0 | 3016.42 | 5917.27 | 21‡ | 77.91 | 0.02 | 10.00 |
| | Hard | New | 11485.33 | 0 | 11485.33 | 22430.84 | 9‡ | 1321.43 | 0.03 | 134.12 |

# 4 Computation Experiments

We illustrate the value of *Distributional MIPLIB* through computational experiments on its MILP distributions. First, we identify distributions that are unexplored in previous work in ML-guided MILP solving and identify potential areas for improvement (subsection 4.1). Furthermore, we propose a novel setting where we learn ML policies from a diverse mix of domains, contrasting with existing work that either trains models on single distributions or completely heterogeneous distributions such as MIPLIB (Table 1) (subsection 4.2). We show that the proposed mixed-domain strategy is particularly effective in data-scarce settings.

Most ML-guided MILP approaches discussed in subsection 2.1 require a computationally expensive data collection procedure before training ML models, as they replace computationally intensive algorithmic components with ML oracles. The output of the expensive algorithmic component (e.g., high-quality neighborhood candidate variables obtained via local branching in LNS) is used as the ground truth in supervised learning. Given that *Distributional MIPLIB* spans many problem

domains and hardness levels, complete benchmarking is beyond the scope of this paper. In this paper, we focus our experiments on Learning to Branch (Learn2Branch) [26], which imitates Strong Branching — a branching rule that effectively reduces the search tree size in B&B but is time-consuming. Learn2Branch encodes a MILP with a variable-constraint bipartite graph, employs a Graph Convolution Network (GCN) to learn variable representations, and trains a policy using imitation learning.

Throughout the experiments, we use SCIP 6.0.1 [28] as the solver[4]. Following existing work [56, 26], we compare the ML methods against Reliability Pseudocost Branching (RPB), a state-of-the-art human-designed branching policy in B&B. We report the mean and standard deviation over 5 seeds for all metrics. We briefly introduce the setup; details on data collection and hyperparameters are deferred to Appendix C.

## 4.1 Learning on Previously Unused Distributions

Table 3: Learn2Branch evaluated on previously unused domains. Note that the solving time differs from Table 2 because results in 2 were evaluated with Gurobi and under different RAM allocations.

| Dist. | Method | Integral | # Opt | Opt Time(s) | NonOpt Gap | # Nodes | Infer Pct(%) | Node Integral |
|---|---|---|---|---|---|---|---|---|
| GISP | SCIP | **118.0** ± **1.5** | **100** | **376.6** ± **4.7** | N/A | 158866.1 ± 1693.4 | N/A | 38596.6 ± 508.0 |
| (Medium) | ML | 139.0 ± 5.8 | 98.0 ± 1.5 | 472.8 ± 16.2 | 0.121 ± 0.007 | **89354.6** ± **2622.5** | 21.2 ± 0.4 | **22636.5** ± **683.3** |
| OTS | SCIP | **20.0** ± **4.7** | **94.4** ± **2.2** | 179.4 ± 5.5 | 0.003 ± 0.002 | **1798.7** ± **242.9** | N/A | **5.8** ± **2.2** |
| (Easy) | ML | 27.9 ± 10.4 | 80.4 ± 7.7 | **129.7** ± **15.2** | 0.003 ± 0.002 | 4073.2 ± 1416.0 | 4.9 ± 1.1 | 20.6 ± 8.2 |
| SRPN | SCIP | **47.0** ± **1.7** | **13.8** ± **0.4** | 54.8 ± 15.1 | **0.152** ± **0.006** | 15420.2 ± 1937.0 | N/A | 1594.2 ± 160.9 |
| (Easy) | ML | 52.3 ± 4.3 | 12.8 ± 0.7 | **41.8** ± **8.4** | 0.16 ± 0.011 | **13003.5** ± **1739.8** | 14.9 ± 1.9 | **1371.6** ± **93.6** |

We evaluate the performance of Learn2Branch on three unused distributions. To our knowledge, GISP has not been used in learning variable branching (Table 1), and OST and SRPN have never been used in any ML-guided methods. We focus on Easy and Medium distributions as learning for branching is typically used on smaller instances in the literature.

**Setup.** We use a train, validation, and test split of 80%, 10%, 10%, respectively. This results in 800 MILP instances used for collecting training data for GISP and OTS and 175 for SRPN-Easy, as SRPN instances are limited. We collect 10 Strong Branching expert samples from each instance. We report the performance metrics described in 3.2 with a time limit of 800s.

**Results and discussions.** As shown in Table 3, the trained ML policy did not outperform SCIP in any of the 3 distributions. We investigate the reason for failure by measuring the number of explored nodes in B&B (# Nodes), the integral of the primal-dual gap with respect to the nodes (Node Integral), and the % of time spent in ML inferences (Infer Pct (%)), which includes feature extraction, forward pass, and ranking. The reason why Learn2Branch did not work well on GISP and SRPN could be the overhead of the ML inference time, as they outperform SCIP on the number of Nodes and Node Integral. For OTS, the reasons why Learn2Branch fails to beat SCIP are less obvious and pose an open research question.

## 4.2 Learning with Mixed Distributions

Collecting expert samples for imitation learning in Learn2Branch is computationally intensive [56]. While collecting a large number of expert samples from a large number of training instances can lead to stronger performance, it could be prohibitively costly. One simple strategy to make the best use of limited data is to pool data and train policies on mixed domains, as opposed to existing work that trains models on a single distribution, distributions from variants of a single problem family [11, 10], or completely heterogeneous distributions such as MIPLIB (Table 1). Empirically, we show that pooling data from a diverse mix of domains achieves better performance when limited training data is used.

**Setup.** We collect samples from training instances from 5 different domains: MIS-Easy, GISP-easy, CFLP-easy, CA-Medium, and SC-Medium. We use the collected data in two different ways. First,

---

[4]We use SCIP in the Experiments as opposed to Gurobi, since Gurobi does not provide needed API for ML-guided branching

Table 4: Performance comparison under two training strategies, evaluated on five domains. Under the first strategy, a separate model is trained for each domain on expert samples collected from instances drawn from homogeneous distributions from the corresponding domain: ML-MIS, ML-GISP, ML-CFLP, ML-CA, and ML-SC. ML-mix5 is trained under the second strategy, where pooled data collected from instances in the 5 domains are used to train one single model. We present results when using different numbers $n$ of training instances per domain: $n = 80$ (left) and $n = 320$ (right).

| Dist. | Policy | Collected samples from 80 instances per domain | | | | Collected samples from 320 instances per domain | | | |
|---|---|---|---|---|---|---|---|---|---|
| | | Integral | # Opt | Opt Time(s) | NonOpt Gap | Integral | # Opt | Opt Time(s) | NonOpt Gap |
| MIS (Easy) | SCIP | $4.412 \pm 0.118$ | $\mathbf{99.0 \pm 0.0}$ | $145.4 \pm 3.9$ | $0.022 \pm 0.004$ | $4.412 \pm 0.118$ | $99.0 \pm 0.0$ | $145.4 \pm 3.9$ | $0.022 \pm 0.004$ |
| | ML-MIS | $5.408 \pm 5.309$ | $82.6 \pm 31.3$ | $140.5 \pm 66.5$ | $\mathbf{0.016 \pm 0.003}$ | $\mathbf{2.434 \pm 0.074}$ | $99.0 \pm 0.6$ | $\mathbf{89.2 \pm 5.1}$ | $\mathbf{0.015 \pm 0.001}$ |
| | ML-mix5 | $\mathbf{2.781 \pm 0.197}$ | $98.0 \pm 1.1$ | $\mathbf{107.3 \pm 13.0}$ | $0.016 \pm 0.004$ | $2.545 \pm 0.107$ | $99.0 \pm 0.6$ | $97.4 \pm 5.4$ | $0.016 \pm 0.003$ |
| GISP (Easy) | SCIP | $12.509 \pm 0.242$ | 100 | $40.0 \pm 0.7$ | N/A | $12.509 \pm 0.242$ | 100 | $40.0 \pm 0.7$ | N/A |
| | ML-GISP | $11.299 \pm 0.885$ | 100 | $41.0 \pm 3.4$ | N/A | $10.700 \pm 0.442$ | 100 | $38.5 \pm 1.6$ | N/A |
| | ML-mix5 | $\mathbf{10.823 \pm 0.383}$ | 100 | $\mathbf{39.1 \pm 2.0}$ | N/A | $\mathbf{10.420 \pm 0.279}$ | 100 | $\mathbf{37.3 \pm 0.8}$ | N/A |
| CFLP (Easy) | SCIP | $0.644 \pm 0.021$ | 100 | $48.5 \pm 0.5$ | N/A | $0.644 \pm 0.021$ | 100 | $48.5 \pm 0.5$ | N/A |
| | ML-CFLP | $0.642 \pm 0.036$ | 100 | $47.8 \pm 3.4$ | N/A | $\mathbf{0.606 \pm 0.028}$ | 100 | $42.4 \pm 1.9$ | N/A |
| | ML-mix5 | $\mathbf{0.638 \pm 0.020}$ | 100 | $\mathbf{46.7 \pm 2.8}$ | N/A | $0.610 \pm 0.021$ | 100 | $\mathbf{42.1 \pm 1.0}$ | N/A |
| CA (Med) | SCIP | $2.347 \pm 0.034$ | $97.2 \pm 0.4$ | $157.4 \pm 4.8$ | $0.009 \pm 0.001$ | $2.347 \pm 0.034$ | $97.2 \pm 0.4$ | $157.4 \pm 4.8$ | $\mathbf{0.009 \pm 0.001}$ |
| | ML-CA | $1.927 \pm 0.063$ | $97.0 \pm 0.0$ | $144.9 \pm 6.2$ | $\mathbf{0.007 \pm 0.001}$ | $1.775 \pm 0.056$ | $98.2 \pm 0.7$ | $\mathbf{136.8 \pm 2.1}$ | $0.009 \pm 0.003$ |
| | ML-mix5 | $\mathbf{1.815 \pm 0.015}$ | $\mathbf{98.2 \pm 0.7}$ | $\mathbf{141.0 \pm 3.2}$ | $0.009 \pm 0.002$ | $1.795 \pm 0.199$ | $\mathbf{98.6 \pm 0.8}$ | $142.1 \pm 12.5$ | $0.011 \pm 0.002$ |
| SC (Med) | SCIP | $6.465 \pm 0.023$ | 100 | $90.3 \pm 0.6$ | N/A | $6.465 \pm 0.023$ | 100 | $90.3 \pm 0.6$ | N/A |
| | ML-SC | $5.602 \pm 0.156$ | 100 | $84.6 \pm 2.2$ | N/A | $4.965 \pm 0.095$ | 100 | $72.5 \pm 1.7$ | N/A |
| | ML-mix5 | $\mathbf{5.362 \pm 0.131}$ | 100 | $\mathbf{79.8 \pm 2.2}$ | N/A | $\mathbf{4.796 \pm 0.104}$ | 100 | $\mathbf{68.4 \pm 1.7}$ | N/A |

we train a separate model for each domain. Second, we pool expert samples collected for all domains and train a single model from the mixed distribution (denoted as ML-mix5). The number of training samples fed into ML-mix5 is five times the first strategy, but the data collection costs for the two strategies aggregated across the 5 domains are the same. We first start with $n = 80$ training instances per domain, which is 10% what we used in 4.1. We then quadruple the number of training instances to $n = 320$. Following 4.1, we collect 10 expert samples per instance. We compare the performance of the two training strategies (single domain vs. mixed domains) on each domain separately.

**Results and Discussions.** As shown in Table 4, when the total number of instances used for data collection is small (80), ML-mix5 outperforms the models trained on homogeneous distributions and SCIP across multiple evaluation metrics for all domains. However, as the number of training instances increases (320), the models trained on a homogeneous distribution outperform ML-mix5 in some domains. This indicates that learning with mixed distributions can improve data collection efficiency in the case when we have a limited budget for data collection (e.g., under time or computational resource constraints), but does not surpass training on homogeneous distributions when training samples can be collected from a larger number of instances. Additionally, Table 4 suggests that when the number of training data points fed into the model is the same, using a training set where the data is drawn from mixed distributions is unlikely to surpass the performance of using a training set where the data is drawn from homogeneous distributions. The performance of ML-mix5 under 80 instances per domain, which was trained with samples collected from 400 training instances in total, did not outperform the separately trained models under 320 instances per domain. This underscores the benefits of having domain-specific distributional datasets as provided in our library.

Table 5: Performance comparison under two training strategies when transferred to different hardness. ML-MIS (trained on *Easy*), ML-SC (trained on *Medium*), and ML-mix5 are the ones presented in Table 4 (under $n = 320$). The time cutoff is 800s, except for *Very hard* distributions where it is 3600s.

| Policy | Integral | # Opt | Opt Time(s) | NonOpt Gap | Integral | # Opt | NonOpt Gap | Infer Pct(%) | Node Integral |
|---|---|---|---|---|---|---|---|---|---|
| | MIS (Medium) | | | | MIS (Very hard) | | | | |
| SCIP | $23.4 \pm 0.1$ | $11.4 \pm 1.0$ | $483.2 \pm 10.5$ | $0.024 \pm 0.0$ | $1479.3 \pm 2.3$ | 0 | $0.393 \pm 0.002$ | N/A | $223.6 \pm 41.7$ |
| ML-MIS | $21.9 \pm 2.4$ | $10.2 \pm 10.4$ | $377.2 \pm 20.0$ | $0.023 \pm 0.003$ | $1461.5 \pm 4.8$ | 0 | $\mathbf{0.390 \pm 0.002}$ | $0.1 \pm 0.0$ | $179.0 \pm 56.4$ |
| ML-mix5 | $\mathbf{16.5 \pm 0.2}$ | $\mathbf{24.2 \pm 3.9}$ | $\mathbf{335.3 \pm 19.6}$ | $\mathbf{0.017 \pm 0.0}$ | $1459.0 \pm 2.4$ | 0 | $\mathbf{0.390 \pm 0.001}$ | $0.1 \pm 0.0$ | $\mathbf{139.4 \pm 42.9}$ |
| | SC (Hard) | | | | SC (Very hard) | | | | |
| SCIP | $53.3 \pm 0.2$ | $35.0 \pm 3.5$ | $378.5 \pm 10.8$ | $0.066 \pm 0.000$ | $\mathbf{767.0 \pm 1.0}$ | 0 | $\mathbf{0.239 \pm 0.001}$ | N/A | $3853.6 \pm 82.6$ |
| ML-SC | $49.6 \pm 0.4$ | $37.8 \pm 1.0$ | $367.1 \pm 7.3$ | $\mathbf{0.062 \pm 0.001}$ | $870.3 \pm 13.6$ | 0 | $0.297 \pm 0.009$ | $9.2 \pm 1.6$ | $\mathbf{2622.8 \pm 579.1}$ |
| ML-mix5 | $\mathbf{48.2 \pm 0.3}$ | $\mathbf{40.2 \pm 0.7}$ | $\mathbf{358.2 \pm 3.3}$ | $\mathbf{0.062 \pm 0.001}$ | $830.3 \pm 19.5$ | 0 | $0.275 \pm 0.010$ | $13.3 \pm 4.7$ | $4051.3 \pm 1734.7$ |

**Transferring to Different Distributions.** We further evaluate the performance of trained models when applied to distributions of different hardness levels from the same domain, for MIS and SC. Table 5 shows that on MIS, ML-mix5 exhibits better generalization to harder instances compared to the model trained on homogeneous distributions, even though ML-mix5 did not outperform ML-MIS at the trained hardness level (Table 4). On SC, again ML-mix5 exhibits better performance than ML-SC on harder distributions of SC, however on the *Very hard* distribution neither is able to outperform SCIP, possibly due to the larger overhead of the GCN inference time on larger instances.

## 5 Potential Research Paths

Below we outline suggestions for potential research paths using *Distributional MIPLIB* to facilitate a significant step-change in the ability to solve hard real-world MILP problems.

**Faster Inference.** Due to computational constraints, prior work has focused on training and testing on relatively small and/or easy MILP distributions. In addition to Learn2Branch, much of the existing work on ML for MILP focuses on replacing an expensive procedure with an ML oracle, such as ML for LNS. Our empirical results highlighted that often the advantage of the ML policy is outweighed by its cost of inference on large MILPs. This calls for investigations of ML model architectures or hardware solutions that specifically target this challenge.

**Synthetic Data Generation.** Synthetic Data Generation (SDG) captures the underlying distribution of a dataset and synthesizes targeted data through a generative process [3]. SDG has been applied to finance [4] and healthcare [38] to address the problem of limited data or preserve the privacy of real data. SDG could also be used to improve ML-based methods for MILPs, as collecting algorithmic decision data from solving instances can be expensive, as discussed in Section 4. There has been existing work that uses data augmentation to generate MILP instances [58, 27, 80, 32] or algorithm decision data inside B&B [56]. *Distributional MIPLIB* could be used to develop theoretical and algorithmic frameworks that generate targeted data forming the same distributions.

**Foundation Model for Combinatorial Optimization.** Deep learning foundation models that leverage vast amounts of data to learn general-purpose representation can adapt to a wide range of downstream tasks, which has drastically transformed the domains of language, vision, and scientific discovery [13]. [54] took a step towards foundation models for MILP by using a LLM-based framework to generate MILPs and training a single model on a diverse set of MILP problems. Moreover, *Distributional MIPLIB* contains MILPs from a wide range of domains and hardness levels, which can be suited for a wide range of tracks (B&B, LNS, and finding primal solutions). Much of the existing works (e.g., learning for backdoors, LNS, and branching) use a common subset of features to learn a representation of MILP variables, which could be unified as a shared latent representation. *Distributional MIPLIB* could be used to develop and train such foundation models for the discrete optimization world.

## 6 Conclusion and Discussion

We introduce *Distributional MIPLIB*, a curated dataset of more than 35 MILP distributions from 13 synthetic and real-world domains, making it a large-scale resource for developing ML-guided MILP solving and comprehensive evaluation. Compared to existing datasets and generators, it provides data in distributional settings which is better suited for ML-guided methods. It provides MILP distributions from a wide range of applications and requires no domain knowledge to access these instances. We intend for the library to continue to grow with domain contributions from the community.

We ran experiments on Learn2Branch focused on variable selection policies in B&B. We identified that in past research only a few distributions/domains were used to assess state of the art, and evaluated the performance of Learn2Branch on unused domains, identifying open challenges. Moreover, we propose to train a Learn2Branch model with mixed distributions and show that this offers advantages in the low-data regime. We also identified potential future directions that can benefit from this library.

We also would like to acknowledge some limitations of our work. Due to computational constraints, we did not experiment with other GNN architectures, with a larger number of samples, or on better GPUs. These could change our empirical conclusions, but do not affect the value of the library.

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

## A  Domain Abbreviations

The abbreviations for the domains are listed in Table 6.

| Abbreviation | Domain | Reference |
|---|---|---|
| CA | Combinatorial Auctions | [53] |
| SC | Set Covering | [5] |
| MIS | Maximum Independent Set | [7] |
| MVC | Minimum Vertex Cover | [23] |
| GISP | Generalized Independent Set Problem | [39, 18] |
| CFLP | Capacitated Facility Location Problem | [20] |
| MK | Multiple Knapsack | [67] |
| MC | Max Cut | [69] |
| CORLAT | Wildlife Management Problem | [19, 30] |
| LB | Load Balancing | [82] |
| IP | Item Placement | [62] |
| MIRP | Maritime Inventory Routing Problem | [65] |
| NNV | Neural Network Verification | [16, 78] |
| OTS | Optimal Transmission Switching | [70] |
| MMCN | Middle-Mile Consolidation Network | [31] |
| SRPN | Seismic-Resilient Pipe Network Planning | [40] |

Table 6: Abbreviation for domains.

# B    Literature Review

**Learning to Branch**    A series of papers have explored learning to branch by imitating the strong branching heuristic, a branching method that results in fewer search tree nodes but is expensive to compute  [46, 60, 2, 6, 26, 33, 64, 56]. The strong branching heuristic computes a score for each branching candidate and these methods either learn to predict the variables' score or learn to rank them according to their scores. For the features and ML models, [46] develop the first ML-based framework for learning to branch using a Support Vector Machine (SVM) with hand-crafted features. [26] extend the framework by using a bipartite graph to encode the MILP and Graph Convolution Networks (GCN) to learn variable representations.

**Learning Backdoors**    Backdoor for MILPs is a small subsets of variables such that a MILP can be solved optimally by branching *only* on the variables in the set [81]. Therefore, identifying backdoors efficiently and effectively can greatly improve the performance of B&B. [24] using ML to predict the most effective backdoor candidates generated by a LP relaxation-based sampling methods. More recently, [15] propose to use a Monte-Carlo tree search method [49] to improve the quality of training data and apply contrastive learning to directly construct backdoors.

**Learning Primal Heuristics**    Primal heuristics refer to routines that find good feasible solutions in a short amount of time [14] and deciding which heuristics to run and when is an important task. These decisions are mostly made by hard-coded frequency rules in MILP solvers, which are static, instance-oblivious, and context-independent. To tackle this challenge, [47] propose a data-driven approach to decide when to execute primal heuristics. [17] derive a data-driven approach for scheduling primal heuristics.

Another line of research is to learn to predict solutions to MILPs. Both [64] and [36] learn to predict optimal solutions to MILPs and fix the values for a subset of variables based on the prediction to get reduced-size MILPs that are faster to solve.

**Large Neighbourhood Search (LNS)**    LNS is a meta-heuristic that can find high quality solutions faster than B&B on large-scale MILP instances but provides no optimality guarantees. It starts with a feasible solution to the MILP and iteratively selects a subset of variables to reoptimize. Local Branching (LB) is a heuristic that finds the variables that lead to the largest improvement over the current solution in each iteration of LNS. But LB is often slow since it needs to solve a MILP of the same size as input. To mitigate this issues, [76] and [41] replace LB with imitation-learned policies. Other ML techniques, such as reinforcement learning (RL), have also been applied to learn destroy heuristics for LNS [75, 84].

**Learning to Cut**    A cutting-plane is a constraint that is valid for feasible integer solutions but cuts into the feasible region of the linear programming (LP) relaxation, thus improving the bound on the optimal solution. Adding cutting planes has been shown to speed up B&B [12, 21]. Modern MILP solvers maintain a cut-pool that includes a large number of cutting planes of a diverse set of classes. The decisions regarding which classes of cutting planes to use, as well as the specific cutting

planes to select from each class, significantly impact solver performance. In recent advancements, [77] introduce a Reinforcement Learning (RL) framework tailored for the Gomory cutting-plane algorithm. Additionally, [44] develop a method to approach cut selection as a learning-to-rank task, while [66] devise a strategy to imitate a lookahead strategy for cut selection.

## C  Experiment Details

We used the Learn2Branch implementation from [26] in our experiments. Their code is publicly available at `https://github.com/ds4dm/learn2branch`.

**Setup.** All experiments in Section 4 were conducted on a cluster with Intel Xeon Gold 6130 CPUs @ 2.10GHz and Nvidia Tesla V100 GPUs. Each method was run with 5 different seeds. For ML-based methods, we trained the model using 5 different seeds and solved the instances using the trained policies that correspond to the 5 training seeds. For the non-ML methods, we used SCIP to solve the instances with 5 different seeds. The results report the mean and standard deviation across these 5 seeds.

**Data Collection.** In the orginal implementation [26], expert samples were collected by sampling from a set of training instances with replacement and solving it with SCIP. They iterated this process until the desired number of expert samples was collected. Therefore, in their implementation, the whole set of training and validation instances was not necessarily used to collect samples. In our implementation, we collected a fixed number of expert samples (10) from each instances, to ensure that all instances in the training set were used.

**Hyperparameters.** We used the same GCN architecture as described in [26] and trained the models in TensorFlow [1]. We used the Adam Optimizer [50] with a batch size of 32 and an initial learning rate of 0.001. In case the when the validation loss does not decrease over a period of 10 epochs, the learning rate was reduced to 20% of its previous value.

## D  License of existing assets

We curated new assets from the following existing assets. The NNV dataset was downloaded from `https://github.com/google-deepmind/deepmind-research/tree/master/neural_mip_solving`, which is available under the terms of the Creative Commons Attribution 4.0 International (CC BY 4.0) license `https://creativecommons.org/licenses/by/4.0/legalcode`. Datasets downloaded from ML4CO (LB, IP, MIRP) are under BSD-3-Clause license `https://github.com/ds4dm/ml4co-competition/blob/main/LICENSE`. CA, SC, MIS, CFLP instances were generated using code from [26], available at `https://github.com/ds4dm/learn2branch?tab=readme-ov-file` under the MIT license `https://github.com/ds4dm/learn2branch?tab=MIT-1-ov-file`.

