# Distributional MIPLIB: Supplemental Materials

**Weimin Huang, Taoan Huang**
University of Southern California

**Aaron Ferber**
Cornell University

**Bistra Dilkina**
University of Southern California

## 1 Access to Data and Code

The MILP instances used for running our experiments can be downloaded from Distributional MIPLIB `https://huggingface.co/datasets/weiminhu/D-MIPLIB`. We used standardized test sets that were established in Distributional MIPLIB for evaluation. Additionally, we documented the MILP formulation and parameters for different distributions on our website `https://sites.google.com/usc.edu/distributional-miplib`, along with links to generators for those available. We open-sourced the generators that we developed at `https://github.com/amf272/Distributional-MIPLIB-Generators`.

We provided the code to obtain the performance metrics of all distributions in this library with Gurobi at `https://github.com/huangwwww/Distributional_MIPLIB_eval`. We used Gurobi (v10.0.3) [2] with 1 hour time limit, on a cluster with Intel Xeon Silver 4116 CPUs @ 2.10GHz, with a RAM allocation of 5G. For SRPN-Hard, MMCN-Very Hard, and OTS-Hard instances, we increased the RAM to 15G, due to memory errors at 5GB.

For the experiments on Learning to Branch in Section 4 in the main paper, we used the Learn2Branch implementation from [1]. Their code is publicly available at `https://github.com/ds4dm/learn2branch`. We modified their code to train ML policies with mixed distributions and to evaluate the methods under more metrics (node integral, and ML inference time). We provided the modified code that we used to run the experiments at `https://github.com/huangwwww/Distributional_MIPLIB_eval`. All the training details, including data split, ML hyperparameters, and evaluation procedure were described in Section 4 and Appendix C in the main paper.

## 2 Impact

### 2.1 Potential positive societal impacts

This paper presents dataset whose goal is to advance the area of ML-guided MILP methods. Our dataset includes real-world optimization problems arising from critical domains such as healthcare, supply chains, and energy planning. There has been a disparity between the development of ML-guided methods and the application of these methods on real-world problems. Studies on these real-world MILP problems in analytics research seldom offer readily available tools for generating the corresponding instances in distributional settings. This is often due to proprietary or private data constraints (e.g., anonymization requirements for location data in routing problems from e-commerce companies) or the lack of parameter specifications (e.g., documenting the distribution under which the generated MILP represent a specific real-world scenarios). In such cases, ML researchers must acquire domain expertise to generate meaningful and realistic real-world instances. As a result, many challenging real-world problems, which could benefit significantly from ML-guided techniques, are overlooked during the development of these approaches. A few papers [3–5] have used instances from real-world problems such as the production packing problem and electric grid optimization [3], but few of the real-world datasets are publicly accessible, making it hard to reproduce the results.

We curate distributions of real-world combinatorial optimization problems from these critical domains, which has the potential to bridge the gap between ML-guided MILP research and real-world applications.

## 2.2 Potential negative societal impacts

We are not aware of negative societal impacts that can be caused by this work.