# OpenReview forum: "Distributional MIPLIB:  a Multi-Domain Library for Advancing ML-Guided MILP Methods"
_NeurIPS.cc/2025/Datasets_and_Benchmarks_Track — Submitted to NeurIPS 2025 Datasets and Benchmarks Track_

### Official Review · Reviewer_Skxi · 2025-06-29

**Rating:** 4
**Confidence:** 1

**Summary:**

This paper introduces Distributional MIPLIB, a new multi-domain library for benchmarking machine learning (ML)-guided methods in Mixed Integer Linear Programming (MILP). The library provides a curated collection of MILP distributions across various problem domains, including synthetic and real-world datasets, categorized by different levels of hardness. The authors demonstrate its utility by evaluating the performance of ML-guided variable branching on these distributions and propose the use of mixed-domain training for ML models, which yields better performance in data-limited settings.

**Dataset Code Accessibility:**

Yes

**Dataset Code Comments:**

The paper explicitly declares the direct reuse of a publicly accessible codebase (Learn2Branch from [26]) and comprehensively documents critical technical implementation details.

**Ethical Considerations:**

No, there are no or only very minor ethics concerns

**Limitations Weaknesses:**

The study's exclusive reliance on GCNs (from Gasse et al. [26])—without exploring alternative graph architectures like GAT or GraphSAGE—undermines claims about the universality of ML-guided branching, as architectural choices may critically impact generalizability. Furthermore, experiments conducted solely on Tesla V100 GPUs overlook how modern hardware (e.g., H100/A100) could drastically reduce inference overhead. This risks overstating the performance gap attributed to 'ML inference costs'  and weakens the justification for prioritizing 'Faster Inference' as a core future direction.

**Strengths Contributions:**

This paper introduces Distributional MIPLIB, a resource designed to facilitate the development of ML-based MILP solvers. The library offers a diverse range of MILP distributions that cater to both small-scale and large-scale optimization problems, which could serve to improve the training and evaluation of ML models used in MILP. Unlike existing libraries, this dataset spans a wide array of real-world problems, helping researchers evaluate ML-guided methods in domains that were previously underexplored in this context. The paper presents clear experiments on the Learn2Branch algorithm, providing insights into how different distributions impact the performance of ML models in solving MILP problems. The comparison between single-domain and mixed-domain training strategies is well-executed, highlighting the practical advantages of using pooled data in constrained settings.

Figures and tables are informative, and the authors effectively use empirical results to justify their claims. The introduction of the mixed-domain strategy for training ML models is particularly novel and demonstrates the potential for more efficient use of limited data, which is an important problem in the field.

---

> ### Author Rebuttal · Authors · 2025-07-31
>
> We sincerely thank the reviewer for the positive feedback and the valuable insights! The primary contribution of our paper is to compile the data and curate the benchmark (obtaining the runtime performance metrics and classifying the distributions). The purpose of our experiments in Section 4 is to highlight the importance of testing the methods on diverse domains and hardness levels, and hence the value of our dataset library in advancing ML-guided MILP solving. The choices of ML architectures and GPUs could change our empirical conclusions, but do not affect the intended point which is that having Distributional MIPLIB as a resource can identify important gaps in current understanding and facilitate research acceleration and novel directions in ML-guided MILP approaches (i.e., highlight the value of the library). We agree that studying different ML architectures and understanding the impact of ML architectures on generalizability and inference time is an important research question. Thank you for the insightful points that hardware choices, neural network architectures and their interplay can all play a role as bottlenecks or as facilitators for ML-guided MILP solving performance. We will make sure to update our discussion of our experimental results with these points. Comprehensive exploration of alternative or new algorithmic/model/hardware ideas are beyond the scope of our dataset paper, but are key research directions we want to highlight as being empowered by the existence of the library we have developed here.

---

> > ### Comment · Reviewer_Skxi · 2025-08-01
> >
> > Thanks for your response. I have decided to maintain my score.

---

### Official Review · Reviewer_ssUu · 2025-06-29

**Rating:** 4
**Confidence:** 4

**Summary:**

This paper introduces *Distributional MIPLIB*, a curated dataset of more than 35 MILP distributions from 13 synthetic and real-world domains. The proposed dataset holds significance in improving the generalization of ML-based approaches for solving MILPs and serves as a potentially valuable resource for developing foundation models in combinatorial optimization.

**Additional Feedback:**

Please see the weakness.

**Dataset Code Accessibility:**

Yes

**Dataset Code Comments:**

This submission is well-documented.

**Ethical Considerations:**

No, there are no or only very minor ethics concerns

**Final Justification:**

Thanks for the author rebuttal. I agree that `Distributional MIPLIB` and `MILP-Evolve` may be complementary, and I believe that exploring training on more diverse, even heterogeneous, data is a promising direction for supporting learning-based MILP solvers in practice. In this sense, despite focusing on homogeneous data, this work offers a certain contribution. I will maintain my evaluation.

**Limitations Weaknesses:**

* I'm curious about the data diversity in your approach and in [1].
  * For example, how does the performance of models trained on Distributional MIPLIB compare to those trained on LLM-generated data, as done in [1]?
  * I'm also interested in any empirical evidence demonstrating whether your benchmark can support the training of MILP-based foundation models.

* There are many dimensions of variation in CO, such as scale, distribution, constraint, objective, etc. While the proposed dataset appears to consider these factors, the paper places particular emphasis on distribution. Could the authors clarify why this dimension is highlighted?

* What do you mean by `heterogeneous` in line 37? Why MIPLIB is less suited for ML-based methods?

* The GitHub repository does not appear to be actively maintained. Do the authors plan to continue supporting the benchmark? If so, what is your future plan?

```
[1] Towards Foundation Models for Mixed Integer Linear Programming. ICLR 2025.
```

**Strengths Contributions:**

* A valuable contribution to learning-based approaches for solving MILPs.
* The paper is well-written, and the benchmark is well-documented.

---

> ### Author Rebuttal · Authors · 2025-07-31
>
> We sincerely thank the reviewer for the positive feedback! We have listed our response as follows
>
> **Data diversity (Limitation 1)**
>
> *Distributional MIPLIB* and the *MILP-Evolve* dataset in [1] are diverse in different ways. *Distributional MIPLIB* is diverse in terms of types of distributions (35+ distributions of varying hardness levels across 13 diverse domains). However, instances within each distribution are homogenous – they have the same problem structure with data parameters (cost vector, coefficient matrix, and right-hand side vector) varying. Each domain in our library is guaranteed to have real-world relevance as the domains are well-specified and well-understood optimization problems. *MILP-Evolve* in [1] is diverse in terms of types of formulations in the set, as they use LLMs to generate new problem formulations by mixing existing formulations. The *MILP-Evolve* set in [1] is closer to a heterogeneous set in this sense, where instances in the same set have different problem structures. *MILP-evolve* can benefit synergistically from our *Distributional MIPLIB* contribution as our 13 domains can be utilized for further LLM-based problem mixing. Hence, the two libraries are complementary, not substitutes of each other.
>
> Curating the *Distributional MIPLIB* dataset as multiple homogenous distributions matches current practice in existing ML-guided MILP solving research and has great value in both benchmarking current methods and advancing research in this field, as an ML policy is typically trained on a single distribution and then evaluated on the same distribution or related distributions of larger size (we elaborate more on this in the next point **Emphasis on distribution**).
>
> We sincerely thank the reviewer for raising the two questions. Training on LLM-generated instances as in [1] and comparing the performance with training using mixed distributions as done in our paper is an interesting new research question. Using *Distributional MIPLIB* to train a MILP foundation model is our next line of research. However, since the primary contribution of our paper is to provide a new and extensive dataset that can enable comprehensive benchmarking of current and future methods, exploring these two questions is out of the scope of our **dataset** paper.
>
>
> **Emphasis on distribution (Limitation 2)**
>
> We focus on distributions because Machine Learning methods typically benefit from a large number of data points drawn from the same distribution (distributional assumptions about the data are at the core of ML methodology, e.g. across images, text, patients, locations, or whatever other object type one may be studying). In the context of Distributional MIPLIB, instances from the same distribution have the same scale and problem structure in terms of constraints and objectives, while the parameters (coefficient matrix, cost vector, and right-hand side vector) vary.
>
> The emphasis on distribution matches what has been done in existing research in ML-guided MILP solving. As shown in Table 1, an ML policy is trained on a single distribution (SD) and then tested on the same distribution that it was trained on (ID) in most existing works. Therefore, we compiled this dataset library focusing on distributions. In Table 2, we documented the publication in which the compiled distribution was first proposed/used in existing ML-guided MILP research.
>
> **Heterogeneous (Limitation 3)**
>
> Instances from the existing MIPLIB are completely heterogeneous – meaning that each instance in MIPLIB has a different problem structure and scale, which do not form a distribution from an ML perspective. This makes it less suitable for ML-based methods because ML typically benefits from a large amount of data from a certain distribution. Training on a large number of MILP instances that have the same structure is important in facilitating the ML model to learn policies tailored to the specific problem structure and is more likely to result in larger improvements in performance. Due to the heterogeneous nature of MIPLIB, there are not enough instances of the same structure that can help the ML model learn a tailored policy, making it less suitable for ML-based methods.
>
> **Github repository and maintenance (Limitation 4)**
>
> We make our library available through multiple channels that are actively maintained: 1) Distributional MIPLIB webpage (link in Supplementary Material) which contains links to precompiled instance datasets and to instance generators ; 2) Github repository (link in Supplementary Material) with generators; 3) HuggingFace repository (link included in the OpenReview submission) hosting all pre-compiled instance datasets, along with instructions for downloading the instances (HuggingFace is the platform of choice for much of ML research). Since we provided pre-compiled MILP instances on HuggingFace, we expect that they would support much of future research and users would use the MILP generators in the Github repository only if more MILP instances from the same distribution or MILP distributions with different hardness levels need to be generated. When new domains are contributed to the Distributional MIPLIB library (as we expect this to be a living and growing resource), the webpage, Github and HuggingFace resources will be appropriately updated to facilitate all our users.

---

> > ### Comment · Reviewer_ssUu · 2025-08-01
> >
> > Thanks for the author rebuttal. I agree that `Distributional MIPLIB` and `MILP-Evolve` may be complementary, and I believe that exploring training on more diverse, even heterogeneous, data is a promising direction for supporting learning-based MILP solvers in practice. In this sense, despite focusing on homogeneous data, this work offers a certain contribution. I will maintain my evaluation.

---

### Official Review · Reviewer_eFrx · 2025-07-03

**Rating:** 4
**Confidence:** 2

**Summary:**

This paper proposes Distributional MIPLIB, providing distributional data (different application domains and hardness level) for ML-based methods and enabling benchmark. Each distribution has a standardized test set (mostly for 100 instances), addressing the heterogeneity and distributional bias in existing MILP datasets. The authors propose to learn branching policies from a mix of distributions,
demonstrating that mixed-distribution training facilitates the generalization ability of ML-based models compared to the training on homogeneous distributions.

**Additional Feedback:**

Please refer to weakness.

**Dataset Code Accessibility:**

Yes

**Ethical Considerations:**

No, there are no or only very minor ethics concerns

**Limitations Weaknesses:**

1. In Section 2, the superiority and differences of the proposed Distributional MIPLIB to existing MIPLIB for ML-guided should be strengthened. It seems to simply integrate existing MIP datasets and divide them to different distributions.
2. The authors list 13 domains and five hardness levels but without justifying why that granularity is proper and optimal. Additional analysis to show this split would be more clear and more useful.
3. How long does it take to generate all these instances or run full-scale mixed-domain training? It is important in reproduce stage, especially for large-scales.
4. The experimental evaluation focuses primarily on Learn2Branch as the ML-guided MILP method, which should Include more ML-guided MILP baselines or heuristics to strengthen the generality and impact of the findings.

**Strengths Contributions:**

1. This paper proposes Distributional MIPLIB for ML-guided MILP methods, with diverse domains and hardness levels (more than 35 MILP distributions from 13 synthetic and real-world domains).
2. Distributional MIPLIB provides data in distributional settings and requires no domain knowledge to access these instances, which is better suited for ML-guided methods.
3. This paper proposes Learn2Branch and conduct experiments to show the superiority of mixed-distribution training for generalizability enhancement.

---

> ### Author Rebuttal · Authors · 2025-07-31
>
> We sincerely thank the reviewer for the positive feedback! We have listed our response as follows
>
> **Contribution compared to existing MIPLIB (Limitation 1)**
>
> We believe that we have made substantial contributions in curating the dataset. We did an extensive literature review on existing ML-guided MILP solving research to obtain the specifications of problem parameters required to re-produce distributions that are consistent with existing work (Table 2) and documented them on our website (website link in Supplementary Material). Detailed information (mathematical formulation and parameter specifications) for each domain can be navigated from the homepage of our website. For domains where the instance generator code is provided in existing work, we run the existing code to compile the distributions as described in their work and generate additional distributions covering more hardness levels for frequently used domains such as MVC. For domains where there is no publicly available generator (GISP, MVC, OTS), we developed the instance generators in order to compile the dataset and open-sourced the generators (Github link in Supplementary Material)
>
> More importantly, we provided MILP domains that have never been used in existing ML-guided MILP research (labelled as *new* in the Dist. Source column in Table 2) and not in the existing *MIPLIB* or in any other publicly available repository. Three real-world domains (OTS, MMCN, and SRPN) were not proposed in distributional settings in existing literature. MILP problems in these three domains were only available as individual and heterogeneous instances in existing literature, as they come from individual case studies. To make them available in distributional settings, we worked with our collaborators to create distributions of MILP instances that are representative of real-world settings by varying domain-specific parameters. For example, the OTS instances are derived from energy networks and wildfire data in the United States. The different distributions (i.e., hardness levels) correspond to energy networks of different sizes and different numbers of days in the planning horizon. Within each distribution, different instances are generated by varying the resource budget parameter and risk threshold parameter, and using wildfire risk simulation data in different months of the year.
>
> **Justification of hardness levels (Limitation 2)**
>
> There is no optimal split in hardness levels. The purpose of the hardness level classification is to capture the characteristics of distributions used in existing ML-guided MILP research. The hardness level is a label that can provide guidance on what MILP distributions to use for benchmarking different ML-guided techniques and conducting further research. For example, ML-guided Large Neighbourhood Search (LNS) is typically used for distributions labeled as *very hard* and has been used in the *very hard* distributions from CA, SC, MIS, and MVC in prior work. When researchers look for different benchmarks for developing and evaluating ML-guided LNS methods in the future, they can look into *very hard* distributions from other domains in *Distributional MIPLIB*.
>
> To provide more guidance on what distributions to use for different methods, we have documented the associated ML-guided MILP solving techniques that have been applied on our website (website link provided in Supplementary Material), for distributions that have been used in existing ML-guided MILP research.
>
> **Time to run full-scale mixed-domain training (Limitation 3)**
>
> The time to run mixed-domain training is about 1 hour in both the 80-samples-per-domain and 320-samples-per domain scenarios, when using Nvidia Tesla V100 GPUs. Following the approach in Learn2Branch, we set a limit of 1000 training epochs with early stopping (training is terminated when there are 20 epochs without improvement). As for the time it takes to generate all these instances, since Distributional MIPLIB is intended as a pre-compiled dataset, we do not expect re-generation of the data. Therefore, we did not record the time it takes to generate the MILP instances.
>
> **Scope of experimental evaluation (Limitation 4)**
>
> The primary contribution of our paper is to compile the data and curate the benchmark (obtaining the runtime performance metrics and classifying the distributions). The purpose of our experiments in Section 4 is to illustrate the value of having such benchmark resources in advancing ML-guided MILP solving by providing two empirical examples of novel research directions that can be unlocked (in particular, multi-domain training and inference time scalability for real-world instances) and support our claim that many more research directions are enabled. Many ML-guided methods are based on supervised learning and require collecting ground truth data, and the computational cost of data collection in combinatorial solving is much higher than in most ML settings as it involves solving NP-hard problems to collect ground truth labels. For example, collecting training data for ML-guided LNS requires 20 hours serially for each single instance. Hence, comprehensive experimentation across multiple ML-guided tasks and approaches evaluated on our benchmarks was prohibitive. We agree that it is important in future research to explore other architectures and techniques, which can provide more insights.

---

> > ### Comment · Reviewer_eFrx · 2025-08-06
> > **Official Comment by Reviewer eFrx**
> >
> > Thanks to the author for the response. I will maintain my score.

---

### Official Review · Reviewer_YkDe · 2025-07-03

**Rating:** 5
**Confidence:** 3

**Summary:**

The work introduces Distributional MIPLIB (d-MIPLIB)—a comprehensive benchmark library of MILP (Mixed-Integer Linear Programming) problem distributions curated from both synthetic and real-world domains. Unlike the traditional MIPLIB, which contains heterogeneous and unstructured instances, d-MIPLIB provides problem-specific distributions that allow for structured and repeatable training and evaluation of machine learning (ML) models for MILP solving.

The authors categorize these distributions across different levels of problem hardness and application domains, and provide pre-generated training and test sets alongside instance generators for many of them. They conduct experiments using Learn2Branch, a known ML branching technique, to highlight how ML performance varies with domain and dataset, and they show the value of training on mixed distributions especially in data-scarce regimes.

**Additional Feedback:**

Questions to the Authors
1. Definition Clarity: What is the precise definition of a "distribution" or "distributional" in the context of Distributional MIPLIB? Does it refer to the sampling distribution of problem parameters, structure, or both?
2. Data Generation Methodology: How are the datasets for OTS, MMCN, and SRPN specifically generated?
3. Extensibility: If we want to introduce a new distribution for a novel MILP problem, what are the recommended guidelines or tools for generating and structuring the dataset?

**Dataset Code Accessibility:**

Yes

**Dataset Code Comments:**

Both the dataset and code are provided and readable.

**Ethical Considerations:**

No, there are no or only very minor ethics concerns

**Final Justification:**

Most of my concerns have been addressed by the authors' detailed response. I will increase my rating to 5 for acceptance.

**Limitations Weaknesses:**

1. Limited Empirical Scope: Experiments are only conducted on Learn2Branch with SCIP and do not explore other architectures or techniques.
2. Ambiguity in Terminology: The definition of a "distribution" or "distributional setting" is implied but not formally specified.
3. Scalability Questions: The paper acknowledges the difficulty of scaling models to very hard or extremely hard instances but doesn’t offer practical solutions yet.

**Strengths Contributions:**

1. Comprehensive Coverage: Includes 35+ distributions across 13 domains, covering both synthetic and real-world problems.
2. Distributional Structuring: Groups instances by problem formulation and hardness level, which is more conducive for ML model training and benchmarking.
3. Facilitates Reproducibility: Provides pre-generated train/test splits and instance generators, helping standardize experimentation.
4. Novel Evaluation Insights: The experiments reveal where ML branching models underperform, indicating areas for further research.
5. Supports Diverse Use Cases: Useful for research in branching, cut selection, LNS, heuristic scheduling, and potentially foundation models.

---

> ### Author Rebuttal · Authors · 2025-07-31
>
> We sincerely thank the reviewer for the positive feedback! We have listed our response as follows
>
> **Definition of distribution (Question 1, Limitation 2)**
>
> In the context of Distributional MIPLIB, a distribution refers to MILPs of the same problem structure with problem parameters sampled from the same distribution. For example, in synthetic graph problems, MIS instances from the same distribution are generated from the formulation of a maximum independent set problem on graphs generated from the same random graph model (e.g., Erdős–Rényi model) with a given number of nodes and a given edge probability. Different graphs generated from the same graph model result in MILP instances with different parameters (cost vector, coefficient matrix, right-hand side vector) that correspond to the same graph distribution.
>
> **Data generation (Question 2)**
>
> We briefly describe the data generation methodology below due to the rebuttal word limit. More details, including the problem description and mathematical formulation for each domain included in d-MIPLIB, are provided in our webpage (link provided in the Supplementary Material). Information on each domain can be navigated from the homepage.
> - OTS:
> The instances are derived from energy networks and wildfire data in the United States. The different distributions (i.e., hardness levels) correspond to energy transmission networks of different sizes and different numbers of days in the planning horizon. Different MILP instances in the same distribution are generated by varying the resource budget parameter (0.0 to 1000.0 in 100.0 increments) and risk threshold parameter (300.0 to 800.0 in 100.0 increments), and using wildfire risk simulation data in different months of the year.
> - MMCN:
> The instances are derived from historical demand data provided by a large U.S.-based e-commerce retailer. The different hardness levels correspond to different middle-mile consolidation networks of different sizes (in terms of number of vendors and facilities). Different MILP instances in the same distribution are created by varying the locations of the vendors and last-mile delivery facilities.
> - SRPN:
> The instances are generated using water network data from the Los Angeles Department of Water and Power. The different distributions correspond to earthquake hazard zones of different sizes that are considered in the problem. Instances in the same distribution are created by sampling subgraphs of the water network in different zones of the city.
>
> **Introducing new distributions (Question 3)**
>
> We sincerely thank the reviewer for raising this important question! The guidelines for introducing new MILP distributions are documented under the “Submit Distribution” tab on our webpage (link provided in the Supplementary Material). We encourage contributors to provide a problem-specific raw dataset with domain-related parameters documented (e.g., network and demand data in routing problems), a description of how instances from the same distribution can be generated (e.g., the range of varying demand in routing problems), and a generator that takes in the raw dataset and outputs MILP instances in \*.lp or \*.mps. If providing domain parameters is not possible due to data privacy issues, providing precompiled instances (\*.lp or \*.mps files) is also acceptable. Additionally, we encourage contributors to run preliminary experiments to obtain the performance metrics and problem instances statistics discussed in Section 3.2 to evaluate the hardness levels of the provided distributions.
>
> **Empirical scope (Limitation 1)**
>
> The primary contribution of our paper is to compile the MILP instances data and curate the benchmark (obtaining the runtime performance metrics and classifying the distributions). The purpose of our experiments in Section 4 is to illustrate the value of having such benchmark resources in advancing ML-guided MILP solving by providing two empirical examples of novel research directions that can be unlocked (in particular, multi-domain training and inference time scalability for real-world instances) and support our claim that many more research directions are enabled. Many ML-guided methods are based on supervised learning and require collecting ground truth data, and the computational cost of data collection in combinatorial solving is much higher than in most ML settings as it involves solving NP-hard problems to collect ground truth labels. For example, collecting training data for ML-guided LNS requires 20 hours serially for each single instance. Hence, comprehensive experimentation across multiple ML-guided tasks and approaches evaluated on our benchmarks was prohibitive. We agree that it is important in future research to explore other architectures and techniques, which can provide more insights.
>
> **Scalability questions (Limitation 3)**
>
> We hypothesize that the challenge in scaling to hard instances lies in the overhead of ML inference time based on our experiments in Section 4. This is a research question that we identified using our benchmark library, and we briefly discussed suggestions for scalability in Section 5, which include faster inference with more efficient ML architecture and hardware. We did not implement these suggestions in this work because new algorithmic/model/architecture ideas are beyond the scope of our dataset paper.

---

> > ### Comment · Reviewer_YkDe · 2025-08-04
> >
> > I would like to thank the authors for their detailed responses to my review comments. Most of my concerns have been addressed. I will increase my rating to 5 for acceptance.

---

### Decision · Program_Chairs · 2025-09-18

**Decision:**

Reject

**Comment:**

The papers proposes a benchmark of mixed-integer linear programming (MILPs) domains and instances for evaluating methods for automatically solving MILPs. The benchmark is wide-ranging, covering problems that the ML community has focused on (that are mostly too easy, e.g., set cover), and also problems that the ML community has mostly ignored (such as maritime inventory routing). A key feature of this benchmark that makes it of high value is its focus on distributions of homogeneous instances, something many MILP benchmarks lack. The paper is well-written and nicely organized, as is the benchmark. The reviewers criticize a lack of evaluations beyond some standard methods, but I think this is not a big deal for the benchmark. I recommend its acceptance.

===== FINAL UPDATE FROM DB Track PCs ====

The final decision for this paper has been taken by the program chairs after consultation with the SACs. All Senior Area Chairs have ranked papers according to the feedback from the AC during the review process. We decided to leave the original meta-review to reflect the opinion of the AC in light of the initial discussions with reviewers and SAC.